# RML Playground: Online Editing, Validation, and Execution for the RDF Mapping Language

Els de Vleeschauwer[1], Arthur Vercruysse[1], Ben De Meester[1] and Pieter Colpaert[1]

[1]*IDLab, Department of Electronics and Information Systems,*
*Ghent University – imec, Technologiepark-Zwijnaarde 122, 9052 Ghent, Belgium*

### Abstract

The RDF Mapping Language (RML) has seen various extensions over the years and was overhauled and modularized by the W3C KGC Community Group. However, this variety of RML versions and varying support for varying features by different mapping engines makes it hard for end-users to know how to create valid mapping rules. We demonstrate RML Playground: a Web application available at https://w3id.org/imec/rml/playground that guides end-users to edit, validate, and execute valid mapping rules in multiple variations. The RML Playground offers (i) a text editor with real-time RDF syntax and SHACL shape validation and autocompletion, powered by the Semantic Web Language Server (SWLS); (ii) a KG construction pipeline using a configurable mapping engine; and (iii) curated example mapping rules illustrating the targeted RML variation. RML Playground is currently deployed in two variations: the latest KGC Community RML version with Burp as mapping engine, and the RML.io-published RML version with RMLMapper as mapping engine. Completely automated using the specifications' ontology and shape descriptions, the RML Playground is easily extensible and maintainable to new RML versions and extensions. Its use of the specifications' documentation allows editors to validate their correctness, and the provision of meaningful hints during the editing process lowers the barrier for using the (new) RML specifications and can be used during training and dissemination activities.

### Keywords

Knowledge Graph Construction, Playground, RDF Mapping Language, Semantic Web Language Server

## 1. Introduction

Since its inception in 2013 [1][1], the RDF Mapping Language (RML)—a mature language to declare mapping rules for constructing Knowledge Graphs from heterogeneous data sources [2]—has seen various extensions [3, 4, 5, 6, 7, 8, 9, 10] and was overhauled and modularized by the W3C KGC Community Group [11].

However, this variety of RML versions and varying support for varying features by different mapping engines makes it hard for end-users to know how to create valid mapping rules: the validity of the rules depends on the RML extensions or variations used and which mapping engine is employed.

In this paper, we demonstrate the RML Playground: a Web application available at https://w3id.org/imec/rml/playground that guides end-users to edit, validate, and execute valid mapping rules in different variations. RML Playground is currently deployed in two variations: the latest KGC Community RML version with Burp as mapping engine, and the RML.io-published RML version with RMLMapper as mapping engine.

## 2. Related Work

Various tools and languages are proposed to help the creation of mapping rules, described using RML or related declarative mapping languages. The work of Duchateau and Debruyne highlights the trade-off between guidance and flexibility between visual tools and textual abstractions in an Integrated

*KGCW'26: 7th International Workshop on Knowledge Graph Construction, May 10th, 2026, Dubrovnik, HR*

✉ els.devleeschauwer@ugent.be (E. de Vleeschauwer); arthur.vercruysse@ugent.be (A. Vercruysse);
ben.demeester@ugent.be (B. De Meester); pieter.colpaert@ugent.be (P. Colpaert)

🆔 0000-0002-8630-3947 (E. de Vleeschauwer); 0000-0003-1586-5122 (A. Vercruysse); 0000-0003-0248-0987 (B. De Meester); 0000-0001-6917-2167 (P. Colpaert)

[1]https://rml.io/specs/rml/

Development Environment (IDE) [12]. **Visual tools** such as RMLEditor [13] and JUMA [14] offer strong guidance for the construction of valid mapping rules but constrain the liberty needed to create complex mapping rules. **Textual abstractions**, such as YARRRML [15] (a YAML-based serialization) and XRM[2] (an abstract syntax aimed to resemble programming languages) offer more flexibility. YARRRML is integrated in Matey [15], an online IDE with an integrated workflow to upload data sources and execute mapping rules, XRM is offered as plugin for existing IDEs, e.g., Eclipse and Visual Studio Code. Their inclusion in an IDE allows to provide guidance through syntax highlighting and autocompletion. Still, their expressivity is limited to the textual abstraction and prevents enriching the mapping rules with metadata that is not necessarily related to RML, e.g., PROV-O statements. GRAPE [12] introduces an IDE which allows users to interlace a textual abstraction of RML—through a Domain-Specific Language (DSL) similar to XML—with additional RDF metadata in one document. GRAPE includes DSLs for Turtle and three modules of the new RML specification for autocompletion. Its compositional approach allows adding DSLs for other RML modules or RDF ontologies, however, GRAPE also depends on the alignment between the DSL abstraction layer and the evolving RML specifications.

Meanwhile, the Semantic Web Language Server (SWLS) leverages the Language Server Protocol (LSP)[3] to bring functionalities such as real-time syntax validation, ontology-aware autocompletion, and SHACL-based diagnostics to any LSP-compatible editor. This allows to integrate SWLS in a broad ecosystem of development environments [16]. SWLS supports multiple serializations (e.g., Turtle and JSON-LD) and allows support for any RDF data model by importing its ontologies and SHACL shapes.

Because RML mapping rules are described in RDF, and is documented with ontologies and SHACL shapes, we can leverage the general-purpose SWLS to support mapping rule editing.

## 3. Architecture

The **RML Playground** builds on and complements the efforts of Matey [15] and SWLS [16], integrating the SWLS into a Web-based editing environment for editing mapping rules.

First, we implemented generic improvements to SWLS to improve support for our use case: *guide* users when writing *valid* mapping rules for a *specific RML variant*. For reliable guidance, the required ontologies and SHACL shapes must be accessible. By default, SWLS's autocompletion includes terms from ontologies available on Linked Open Vocabularies[4]. Additionally, SWLS allowed to import ontologies and SHACL shapes through prefix declarations and `owl:import` statements inside the document that is being edited. However, this required such RML-specific duplicate statements for each mapping document. For this, SWLS was extended with a configurable list of ontologies and SHACL shapes URLs, managed on application level. To secure valid mapping rules, autocompletion for RML terms must always fit the semantic context. By default, SWLS orders properties for autocompletion in such a way that properties with the correct domain are shown first, and visually distinguishes them from other properties. SWLS has been extended with an option to restrict the autocompletion for certain namespaces to *only* properties with the correct domain. This keeps the focus on the targeted RML variant, as the user no longer gets suggestions related to other RML variants.

Next, we enhanced the JavaScript codebase of Matey by integrating the Monaco editor[5]. As Monaco supports the Languages Server Protocol, this allows to easily integrate SWSL. To facilitate the SWLS integration into the JavaScript codebase, this integration code was released as npm package:`swls-monaco`[6]. Finally, we enabled configurability of a specific deployment, customizing the playground to the targeted RML variant: select a specific mapping engine, import specific ontologies and SHACL shapes, and include curated example mapping rules.

The code of RML Playground is available online at https://github.com/RMLio/playground/v1.0.0 under the permissive MIT license. The RML Playground is deployed as a static site and runs entirely in

---

[2]https://zazuko.com/products/expressive-rdf-mapper/
[3]https://microsoft.github.io/language-server-protocol/
[4]https://lov.linkeddata.es/dataset/lov/
[5]https://microsoft.github.io/monaco-editor/
[6]https://www.npmjs.com/package/swls-monaco

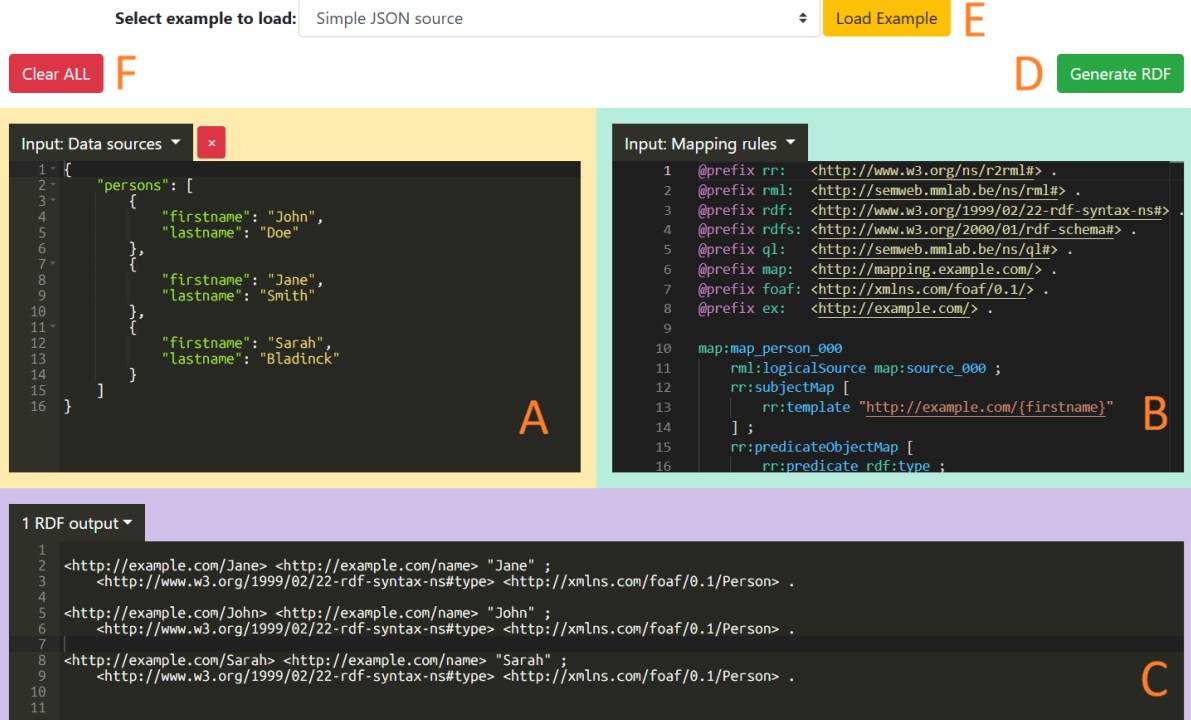

**Figure 1:** The GUI of RML Playground consists of three panels: editing source data (top-left, A), editing RML mapping rules (top-right, B), and displaying constructed RDF data after executing the mapping pipeline (bottom, C).

the browser. Only the mapping engine must be provided as a separate API endpoint; all editing and rule handling are client-side.

RML Playground provides user guidance in three complementary ways for any configured RML variation: (i) the text editor provides real-time RDF and RML syntax checking: RDF syntax through native SWSL functionality, and RML syntax through preconfigured SHACL shape-based validation and autocompletion; (ii) a preconfigured KG construction pipeline that is compliant with the configured SHACL shape(s); and (iii) curated example mapping rules illustrating the targeted RML variation.

To demonstrate it general applicability, we have currently deployed RML Playground in two variations. RML.kgc Playground[7] includes examples illustrating each module of the latest KGC Community RML version and uses its reference implementation Burp[8] [17] as mapping engine in its mapping pipeline. RML.io Playground[9] targets the RML version[10] published on RML.io and its extensions RML-FNML, RML-Target, incRML, RML HTTPRequest and RML Dynamic Target[11], and examples illustrating those specifications. and uses RMLMapper[12] as mapping engine in its mapping pipeline. The configurable set-up allows us to version these deployment so that we can later support different RML mapping engine versions in parallel.

## 4. Demo

The RML Playground's Graphical User Interface (GUI) consists of three panels (Figure 1). The top left panel (Figure 1 A) is the editing area for data sources from which RDF data is generated. Multiple data

---

[7]https://w3id.org/imec/rml/playground/kgc/burp/v0

[8]https://https://github.com/kg-construct/BURP/tree/kgc-2025-conformance

[9]https://w3id.org/imec/rml/playground/rmlio/rmlmapper/v8.1.0

[10]https://rml.io/specs/rml/

[11]https://fno.io/rml/, rml-fnmlhttps://rml.io/specs/rml-target/, https://knowledgeonwebscale.github.io/incrml-spec/, https://rml.io/specs/access/httprequest/, and https://rml.io/specs/target/dynamictarget/, respectively

[12]https://github.com/RMLio/rmlmapper-java/, v8.1.0

sources can be added through the use of a drop down menu with data in different formats, such as CSV, JSON, and XML. A screencast demonstrating all features of the RML Playground is available at https://w3id.org/imec/rml/playground/screencast/2026-kgcw.

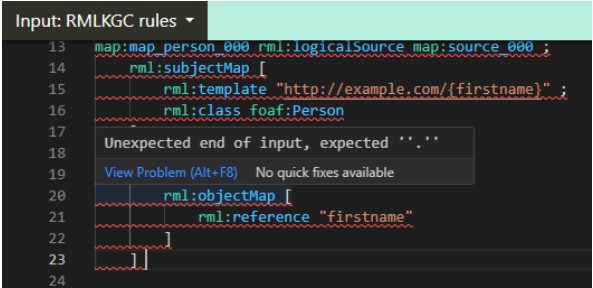

(a) Turtle syntax validation

(b) Auto completion based on ontology descriptions

(c) Shape validation based on SHACL shapes

(d) Error thrown by the mapping engine

**Figure 2:** User guidances: Turtle syntax validation, autocompletion, shape validation, and error descriptions.

The top right panel (Figure 1 B) is the editing area for mapping rules. When writing mapping rules, the user is guided by real-time Turtle syntax validation (Figure 2a), autocompletion (Figure 2b), and SHACL shape validation (Figure 2c). The Turtle syntax validation marks syntax problems such as unexpected characters or undefined prefixes, adding textual explanations on hover. Additionally, consistent formatting and syntax highlighting improves readability of the mapping rules. The autocompletion of RML terms is restricted to properties with a compatible domain. The autocompletion is not restricted to RML terms: terms are suggested for any prefix included in Linked Open Vocabularies[13], and all prefixes declared at the top of the mapping document (on the condition that the referred resources are online accessible), as help for the user when mapping source data to RDF terms or when extending mapping rules with metadata. However, RML terms from other RML variants, not targeted by the deployed playground, will not be proposed to the user. Visual markings indicate a violation of the SHACL shapes of the targeted RML variant. On hover, the user can read which SHACL rule is unsatisfied and the related SHACL message (if this is specified in the underlying SHACL shape).

The bottom panel (Figure 1 C) displays constructed RDF data after executing the mapping pipeline by clicking the bottom "Generate RDF" (Figure 1 D). When multiples targets are defined in the mapping rules, multiple datasets will be generated. The number of generated RDF datasets is displayed in the title of the bottom panel. Clicking this title, a dropdown list appears, allowing the user to switch between the generated datasets or to download them. Any error messages reported by the underlying mapping engine are displayed as a pop-up notification to the user, as aid to correct the mapping document.

The user can load curated examples, selecting them from the example dropdown list and clicking the "Load example" button (Figure 1 E). Each example includes one or more datasets and a mapping document. Finally, the "Clear all" (Figure 1 F) button allow a fresh restart with blank editing panels.

---

[13]https://lov.linkeddata.es/dataset/lov/vocabs

## 5. Conclusion

The demonstrated RML Playground allows end-users to edit, validate and execute RML mapping rules in different variations. The direct integration of ontologies and SHACL shapes enables consistent alignment with the targeted RML specification, avoiding the dependency on an RML-specific abstraction layer. Completely automated using the specifications' ontology and shape descriptions, the RML Playground is easily extensible and maintainable to new and updated RML versions, extensions, and mapping engines. Its use of the specifications' documentation allows editors to validate their correctness, and the provision of meaningful hints during the editing process lowers the barrier for using the (new) RML specifications and can be used during training and dissemination activities.

## Acknowledgments

The described research activities were supported by SolidLab Vlaanderen (Flemish Government, EWI and RRF project VV023/10), the European Union's Horizon Europe research and innovation program under grant agreement no. 101058682 (Onto-DESIDE), and the imec ICON project Solid4Media (Agentschap Innoveren en Ondernemen project nr. HBC.2022.0770).

## Declaration on Generative AI

The authors used Copilot for grammar and spelling check, reviewed and edited the content as needed, and take full responsibility for the publication's content.

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
