# OpenReview forum: "RML Playground: Online Editing, Validation, and Execution for the RDF Mapping Language"
_eswc-conferences.org/ESWC/2026/Workshop/KGCW — KGCW 2026_

### Official Review · ~Jakub_Duchateau1 · 2026-03-31
**Interesting tool to demo; needs to clarify the attempted problem**

**Rating:** 6
**Confidence:** 4

**Review:**

This demo paper presents **RML Playground**, an evolution of Matey IDE, with the ability to choose the RML variant (rml.io or kgc community) via the integration of RMLMapper and BURP. They start from the problem that it is hard for end-users to create valid mapping rules. The text editor was replaced by Monaco and integrates the Semantic Web Language Server to provide turtle syntax help, ontology and/or shapes completion preconfigured for the right RML variant and SHACL-based validation. The SWLS itself was improved for this use case with the support for restricted namespaces and implicit ontology and shapes predefined at the application level. Turtle is used in both variants as the mapping syntax, whereas Matey originally used YARRRML.

While highly relevant to the KGC community for educational and training purposes, the paper currently lacks a clearly defined scientific or technical contribution and includes no user evaluation. I rate it barely above the threshold currently, but I would rate it higher if the core research or engineering problem were made explicit.

## Strengths

- The tool is *open-source* and available *online*, removing the setup and installation steps for users wanting to experiment with RML.
- The integrated architecture (with Matey, Monaco, SWLS and various RML processors) is a useful tool for the community.
- The inclusion of a preconfigured environment and curated examples makes it a handy resource for training, dissemination and sandboxed mapping creation.
- The fact that for an RML variant we are given compatible resources is also a strength.

## Weaknesses

- To help strengthen the article's **contribution**, could you elaborate on the research question or the gap (i.e., in tooling) you wanted to address? If your goal was not to answer a research question but to achieve an engineering challenge, what does this integrated architecture provide beyond the sum of its individual (enhanced) components?
- A **human evaluation** would be a great addition. As it is similar to Matey, I expect it could be transferred, but I did not find an evaluation of Matey either. Any kind of user, expert, end-user testing or heuristic testing would be valuable.

## Minor Comments & Clarifications

- The use of the term "end-users" could be ambiguous in this context. In software development literature, "end-user development" usually implies individuals who are not professional developers. It could be clarified exactly who the target user is or left unspecified.
- The paper uses "RML variants" and "variations" interchangeably. It would be helpful to only use one of them. Especially because we also have RML modules that we could swap in and change the meaning completely. :)
- And on that topic, what about different RML modules? Could the SWLS completion be restricted to a subset of modules only?
- In conclusion, "end-users" – who is it referring to? In the context of development/interfaces, it could be confused with end-user development tools, and so people who are not professional software developers, which I am not sure will fit well.

---

### Official Review · ~Mario_Scrocca1 · 2026-04-02
**Demonstration paper introducing a potentially valuable tool, presentation could be improved**

**Rating:** 6
**Confidence:** 5

**Review:**

This demo paper presents RML Playground, a browser‑based environment for creating, validating, and executing RML mapping rules across multiple configurations. The system integrates the Semantic Web Language Server (SWLS) and is configurable for different versions/extensions of the RML specification to provide real‑time syntax validation, SHACL‑based constraint checking, ontology‑aware autocompletion. The tool also presents an execution interface configurable with different RML mapping processors.  The demonstrated version shows two different configurations and also offers curated examples. I appreciate the inclusion of a screencast of the tool's features, as it could help practitioners reading the paper.

The demonstration is clearly relevant to the workshop, however, I have some comments on the current version of the paper:
- The authors could better discuss the expected target users (e.g., novice users learning how to write RML mapping rules).
- The concept of “RML variation” is confusing in my opinion. For new users (likely a core target audience of the tool), this wording unintentionally suggests that there are competing specifications and that the value of the tool is the possibility of supporting both. I would recommend reframing the terminology around “versions” and “extensions” of the RML specification. In general, I believe the "selling point" should be that the tool can evolve to support different versions and extensions of RML by combining: the relevant ontology, the SHACL shapes, and the chosen mapping engine.
- Clarify more explicitly the distinction between the generic tool and specific deployed instances for demonstration. At times, it is unclear what the generic functionality of RML Playground is and what belongs to a specific deployment configuration for the demonstration. When reading statements like “the mapping engine must be provided as an API endpoint,” I initially wondered whether this applies to all instances, including those already deployed.
- Regarding configuration with other mapping processors, it is not clear whether there is a documented API specification for integrating them with the Playground.
- Minor issues: The footnote 9 link seems broken and should be corrected.

Additional comments on the tool:
1. In my experience, developers prefer working within existing IDEs (e.g., VS Code), so I strongly encourage the authors to consider releasing the SWLS customizations as a VS Code extension, allowing users to benefit from the same validation and autocompletion features without needing to move to a separate tool.
2. YARRRML support would be extremely valuable. Although RML Playground builds on Matey, the current tool focuses exclusively on RML in its RDF serialisation. In practice, many users (including myself) author mappings primarily in YARRRML. Extending the Playground to support YARRRML with similar validation and guidance features would be a major improvement and significantly broaden its usefulness.

Despite these comments, I believe the paper introduces a tool that could already be valuable to certain users (e.g., novice users wrt RML), and its demonstration during the workshop can help gather additional feedback from the KGC community for its future development.

---

### Official Review · ~Oscar_Corcho1 · 2026-04-06
**Demo paper clearly related to the topic of the workshop**

**Rating:** 6
**Confidence:** 4

**Review:**

This demo paper is clearly related to the topics of the workshop and shows some relevant advances over the previous versions of similar tools, such as Matey. The workshop can benefit from the presentation of this tool and the discussions of the community on strengths and weaknesses, and potential new functionalities based on the experiences of those in the audience.

Even though the AI declaration states that GenAI has been used to polish grammar and typos, I have still found many while reading, so I would recomment a second pass into it. Note that the GitHub link to the released version does not seem to work, although the repository is clearly there.

The paper itself is ok for a demo paper, describing both the design decisions taken (some of which may be a bit hard to understand by those who are not completely aware of architectures and underlying technologies (e.g., SWLS)) and the functionalities that are available for users of the tool.

My only considerations, as an ocassional user of the previous Matey, are the following:
- I would encourage the support for YARRRML, even though there are some limitations in this kind of approach, as discussed by the authors, in terms of not being able to provide additional annotations (e.g., PROV-O).

---

### Decision · Program_Chairs · 2026-04-09

**Decision:**

Accept

**Comment:**

This paper has been selected for presentation at the KGC workshop. We strongly encourage the authors to consider the reviews whilst revising the paper. Camera-ready instructions will soon follow.